# Involvement of P2Y_1_, P2Y_6_, A_1_ and A_2A_ Receptors in the Purinergic Inhibition of NMDA-Evoked Noradrenaline Release in the Rat Brain Cortex

**DOI:** 10.3390/cells12131690

**Published:** 2023-06-22

**Authors:** Clara Quintas, Jorge Gonçalves, Glória Queiroz

**Affiliations:** Mechanistic Pharmacology and Pharmacotherapy Unit, UCIBIO-i4HB, Laboratory of Pharmacology, Department of Drug Sciences, Faculty of Pharmacy, University of Porto, 4050-313 Porto, Portugal; claraquintas@ff.up.pt (C.Q.); gloria@ff.up.pt (G.Q.)

**Keywords:** NMDA receptors, noradrenaline release, P2Y_1_ receptors, P2Y_6_ receptors, A_1_ receptors, A_2A_ receptors, rat cortical slices

## Abstract

In the cerebral cortex, glutamate activates NMDA receptors (NMDARs), localized in noradrenergic neurons, inducing noradrenaline release that may have a permissive effect on glutamatergic transmission, and therefore, on the modulation of long-term plasticity. ATP is co-released with noradrenaline, and with its metabolites (ADP and adenosine) is involved in the purinergic modulation of electrically-evoked noradrenaline release. However, it is not known if noradrenaline release evoked by activation of NMDARs is also under purinergic modulation. The present study aimed to investigate and to characterize the purinergic modulation of noradrenaline release evoked by NMDARs. Stimulation of rat cortical slices with 30 µM NMDA increased noradrenaline release, which was inhibited by ATP upon metabolization into ADP and adenosine and by the selective agonists of A_1_ and A_2A_ receptors, CPA and CGS2680, respectively. It was also inhibited by UTP and UDP, which are mainly released under pathophysiological situations. Characterization of the effects mediated by these compounds indicated the involvement of P2Y_1_, P2Y_6_, A_1_ and A_2A_ receptors. It is concluded that, in the rat brain cortex, NMDA-evoked noradrenaline release is modulated by several purinergic receptors that may represent a relevant mechanism to regulate the permissive effect of noradrenaline on NMDA-induced neuroplasticity.

## 1. Introduction

In the central nervous system (CNS), noradrenergic neurons comprise different subpopulations. These differ in their anatomical location, efferent projection pattern, connectivity and function [1]. The locus coeruleus, located in the brain stem, is the principal origin of noradrenergic neurons that project to the cerebral cortex [2]. In the cerebral cortex, noradrenaline is involved in the modulation of behavior, such as the sleep/wake states, mood and stress response [3,4,5,6,7]. Additionally, it influences the modulation of long-term plasticity at glutamatergic synapses [8], where it is thought to have a permissive effect on NMDA-mediated glutamatergic transmission [8,9,10].

Glutamate is the main excitatory neurotransmitter in the CNS and exerts its effects through the activation of three membrane ionotropic receptors, named NMDA, AMPA and kainate and eight subtypes of metabotropic receptors (mGLU_1–8_). NMDA receptors (NMDARs) exist as diverse subtypes formed by variation in assembly of seven subunits (GluN1, GluN2A-D and GluN3A-B) into tetrameric receptor complexes that allow the influx of Ca^2+^ and Na^+^ and the efflux of K^+^ [11]. Several studies have revealed that NMDARs formed by combination of the GluN1 and GluN2 subunits are the major mediators of neuronal plasticity [9,10]. However, they are also involved in the pathophysiology of brain disorders leading to neuronal loss [12]. NMDARs are mostly localized in cell bodies, in dendritic spines, but also in presynaptic nerve terminals, inducing or modulating the release of several neurotransmitters [13], including noradrenaline release in the cerebral cortex [14,15,16].

The noradrenergic neurons that project to the cortex also release ATP as a co-transmitter [17]. Other nucleotides, such as UTP and its metabolites, may also be released from neurons and glial cells and modulate noradrenergic transmission. In fact, in rat brain cortex, these nucleotides, as well as adenosine, are involved in the modulation of electrically-evoked noradrenaline release by activation of several purinergic receptors [18,19]. Purinergic receptors have a widespread expression in the CNS and may be activated by ATP and other nucleotides released from neurons and glial cells [20]. They include the P1 receptors, adenosine A_1_, A_2A_, A_2B_ and A_3_ subtypes [21], eight subtypes of nucleotide metabotropic P2Y receptors, P2Y_1,2,4,6,11–14_ [22] and seven subunits P2X_1–7_, which form homomeric or heterometric ionotropic P2X receptors channels [23]. 

Although electrically-evoked neurotransmitter release is a model that mimics the activation of wider neural circuits, it does not enable us to distinguish one in particular, such as the glutamatergic circuit. Therefore, it is not known whether noradrenaline release evoked by activation of NMDARs is also modulated by purinergic receptors.

The present study aimed to characterize the purinergic modulation of noradrenaline release evoked by the activation of NMDARs. Understanding the influence and mechanisms by which purinergic receptors control the release of noradrenaline evoked by NMDARs in the brain cortex may open new lines of intervention in CNS pathologies that course with a dysregulation of noradrenergic transmission, such as in mood disorders, cognition and the progression of neurodegenerative diseases such as Parkinson’s disease and Alzheimer’s disease [24,25] or dysregulation of glutamatergic transmission, which occurs in ischemia, epilepsy, mood disorders and neurodegenerative diseases [26,27,28,29].

## 2. Materials and Methods

### 2.1. Drugs

The following drugs were used: levo-[ring-2,5,6-^3^H]-noradrenaline ([^3^H]-NA), specific activity 1.65 TBq/mmol, was from DuPont NEN (Garal, Lisbon, Portugal); adenosine-5’-diphosphate disodium salt (ADP); adenosine 5′-triphosphate disodium salt (ATP); adenosine deaminase type VI (ADA; EC 3.5.4.4), D(-)-2-amino-5-phosphonopentanoic acid (D-AP5); bisindolylmaleimide XI (Ro 32-0432); N^6^-cyclopentyladenosine (CPA); 2-p-(2-carboxyethyl) phenethylamino-5′-N-ethylcarboxamidoadenosine hydrocloride (CGS 21680); 8-cyclopentyl-1,3-dipropylxanthine (DPCPX); 6,7-dinitroquinoxaline-2,3(1H,4H)-dione (DNQX); dizocilpine (MK-801); 6-N,N-diethyl-β-γ-dibromomethylene-D-adenosine-5′-triphosphate trisodium salt hydrate (ARL 67156); α,β-methyleneadenosine 5’-diphosphate lithium salt (AOPCP); N-[2-(p-bromocinnamylamino)ethyl]-5-isoquinolinesulfonamide dihydrochloride (H-89); tetrodotoxin (TTX); uridine 5′-triphosphate trisodium salt (UTP); uridine 5′-diphosphate disodium salt (UDP); N-[2-[N-(4-chlorocinnamyl)-N-methylaminomethyl]phenyl]-N-(2-hydroxyethyl)-4-methoxybenzenesulfonamide phosphate salt (KN-93); N-methyl-D-aspartate (NMDA); 7-(2-phenylethyl)-5-amino-2-(2-furyl)-pyrazolo-[4,3-e]-1,2,4-triazolo[1,5-c]pyrimidine (SCH 58261); tetrodotoxin (TTX); 1-[6-[((17β)-3-methoxyestra-1,3,5[10]-trien-17-yl)amino]hexyl]-1H-pyrrole-2,5-dione (U-73122) and 1-[6-[((17β)-3-methoxyestra-1,3,5[10]-trien-17-yl)amino]hexyl]-2,5-pyrrolidinedione (U-73343). Uridine-5′-diphosphate disodium salt (UDP) and uridine-5′-triphosphate trisodium salt (UTP) were from Sigma (Lisbon, Portugal) and N,N″-1,4-butanediylbis-N’-(3-isothiocyanatophenyl)thiourea (MRS 2578), 2-[(2-chloro-5-nitrophenyl)azo]-5-hydroxy-6-methyl-3-[(phosphonooxy)methyl]-4-pyridinecarboxaldehyde disodium salt (MRS 2211), 2-(propylthio)adenosine 5′-O-(β,γ-difluoromethylene) triphosphate tetrasodium salt (AR-C 66096) and (1R*,2S*)-4-[2-iodo-6-(methylamino)-9H-purin-9-yl]-2-(phosphonooxy)-bicyclo[3.1.0]hexane-1-methanol dihydrogen phosphate ester tetraammonium salt (MRS 2500) were from Tocris (Bristol, UK).

Stock solutions of drugs were prepared with DMSO or distilled water and kept at −20 °C. Solutions of drugs used in the experiments were prepared from aliquots of the stock solutions that were diluted in buffer immediately before use. Solvent was added to the superfusion medium in parallel control experiments.

### 2.2. Preparation of Slices from Rat Brain Cortex

Animal handling and experiments were conducted according to the guidelines of the European Union Directive 2010/63/EU and approved by the local (ORBEA-ICBAS-UP) and national (DGAV) competent authorities. Adult male Wistar rats (Charles River, Barcelona, Spain) were kept at a constant temperature (21 °C) and a regular light (06.30–19.30 h)/dark (19.30–06.30 h) cycle, with food and water ad libitum.

After euthanasia, the brains of approximately fifty animals were quickly removed and chilled and other parts of the animals were used in different research projects. Transverse slices were cut from the occipital-parietal cortex. From each brain, the first superficial slice was discarded, and the subsequent three slices from each hemisphere (six slices per animal) of 400 µm (Microtome Leica VT1000S, Nussloch, Germany) were cut and incubated for 45 min in a warmed (37 °C) and gassed (95% O_2_ and 5% CO_2_) Krebs solution with the following composition (mM): NaCl 118, KCl 4.7, CaCl_2_ 2.5, MgSO_4_ 1.2, NaH_2_PO_4_ 1.2, NaHCO_3_ 25, glucose 11, ascorbic acid 0.3 and disodium EDTA 0.03.

### 2.3. Experiments of [^3^H]-Noradrenaline Release

The procedures used to label the slices with [^3^H]-noradrenaline ([^3^H]-NA) and to estimate changes in chemically or electrically-evoked tritium release as an indicator of neuronal noradrenaline release from cortical brain slices have been previously described [14,19]. Briefly, occipital-parietal cortical brain slices were incubated in 2 mL of Krebs solution containing 0.1 µM [^3^H]-NA (specific activity of 1.65 TBq/mmol) for 15 min. After the incubation period, the slices were washed for 10 min and one slice was transferred to each superfusion chamber, where they were held by a polypropylene mesh between platinum electrodes placed 7 mm apart. Each chamber was then superfused at a constant flow rate of 0.6 mL/min with solvent, a modified Krebs solution, containing 0 mM Mg^2+^ (MgSO_4_ was replaced by an equimolar concentration of NaCl) or the drug treatments at 37 °C, and oxygenated with 95% O_2_ and 5% CO_2_.

A stimulator (Hugo Sach Elektronik, Type 215, Hugstetten, Germany), operating in the constant current mode, was used for a primer electrical stimulation period, which consisted of square-wave pulses (1 ms width; 50 mA current strength; voltage drop of 18 V per chamber; 1 Hz, 20 pulses) applied at t = 30 min (S_0_), t = 0 min being the onset of superfusion, and was not used for determination of [^3^H]-NA release. Two other stimulation periods were applied at t = 60 min (S_1_) and t = 105 (S_2_), consisting of the superfusion with a solution of NMDA 30 µM for 2 min. Superfusate samples were collected at 5 min intervals, from t = 55 min onwards until t = 130 min. At the end of the experiments, the tritium remaining in the slices was extracted with 0.2 M NaOH overnight at 4 °C and the tritium content was determined in the collected superfusate samples and in tissue extracts by scintillation spectrometry (Beckman LS 6500, Beckman Instruments, Fullerton, CA, USA). In some experiments Mg^2+^ was added or Ca^2+^ were omitted from the superfusion medium 30 min before S_2_ until the end of the experiment. Agonists were added to the superfusion medium 10 min before S_2_ and were kept until the end of the stimulation period. Antagonists were added to the superfusion medium 20 min before S_2_, being kept until the end of the experiment. In some experiments, DPCPX (a selective A_1_ receptor antagonist) was present throughout superfusion.

Tritium release was estimated as a fraction of the slice tritium content at the onset of the respective collection period (fractional rate of tritium release.min^−1^); b_1_ and b_2_ were the fractional rate of tritium release in the 5 min period before S_1_ and S_2_, respectively. Drug effects on basal tritium release were evaluated as ratios b_2_/b_1_. Tritium release evoked by NMDA 30 µM during 2 min (NMDA-evoked [^3^H]-NA release) was estimated by subtracting basal tritium release from total tritium release observed during and in the 10 min period subsequent to each stimulation period and was expressed as percentage of the total tritium present in the slice at the onset of respective stimulation period. Effects of drugs added after S_1_ were evaluated as ratios of the release elicited by S_2_ and the release elicited by S_1_ (S_2_/S_1_) and were expressed as percentage of change (inhibition or facilitation) from the mean ratio obtained with the respective control.

### 2.4. Statistical Analysis

Data are expressed as means ± standard deviation of the mean (SD) from n number of experiments. Statistical analysis of the effect of drugs on basal tritium release and NMDA-evoked [^3^H]-NA release was carried out using the unpaired Student’s *t*-test or one-way analysis of variance (ANOVA) followed by Dunnett’s multiple comparison test. *p* values lower than 0.05 were considered to indicate significant differences.

## 3. Results

### 3.1. Characterization of NMDA-Evoked [^3^H]-NA Release

Stimulation of rat cortical slices with 30 µM NMDA increased the fractional rate of tritium release (Figure 1). The [^3^H]-NA release evoked by 30 µM NMDA during 2 min (S_1_) was 2.183 ± 0.697% (*n* = 110) of total slice tritium content. In the absence of any drug or solvent, the ratio S_2_/S_1_ was 0.90 ± 0.09 (*n* = 110). None of the drugs tested after S_1_ or their solvents modified basal tritium release, except the omission of Ca^2+^ from the medium, which promoted an increase of basal tritium release. When the adenosine A_1_ antagonist DPCPX was present throughout superfusion, it did not change either basal tritium release or NMDA-evoked [^3^H]-NA release (not shown).

When slices were superfused with medium containing 1.2 mM Mg^2+^, [^3^H]-NA release was abolished (Figure 1 and Figure 2), confirming the involvement of NMDA receptors. Omission of Ca^2+^ from the superfusion medium and blockade of voltage-sensitive Na^+^ channels (VSNaC) with TTX (0.3 µM) also abolished NMDA-evoked [^3^H]-NA release (Figure 2). Under these experimental conditions, there was no contribution of other ionotropic glutamate receptors, since the AMPA and kainate receptor antagonist DNQX (10 µM) did not change NMDA-evoked [^3^H]-NA release, which was reduced by the selective NMDA receptor antagonist D-AP5 (50 µM) and abolished by the selective and non-competitive NMDA receptor antagonist MK-801 (50 µM; Figure 2). These results demonstrate that, under these conditions, NMDA receptors evoke the release of [^3^H]-NA by an action potential-dependent exocytosis.

### 3.2. Effect of Nucleotides on NMDA-Evoked [^3^H]-NA Release 

In previous studies it was shown that in rat brain cortex, multiple purinergic receptors were involved in the modulation of [^3^H]-NA release evoked by electrical stimulation [18,19]. In this study, we investigated the influence of several nucleotides, with different selectivity for P2 receptors, on [^3^H]-NA release evoked by NMDA receptors activation: ATP, which is an agonist of both P2X and P2Y receptors; ADP that has higher affinity for the P2Y_1,12,13_ receptor subtypes; UTP that preferentially activates the P2Y_2,4_ receptor subtypes and UDP, which is selective for the P2Y_6_ receptors [30]. All the nucleotides tested inhibited NMDA-evoked [^3^H]-NA release (Table 1).

### 3.3. Characterization of P2 Receptors Activated by Adenine Nucleotides

The effects of ATP and ADP were characterized in more detail, to identify the purinergic receptors involved in the inhibition of NMDA-evoked [^3^H]-NA release. The inhibitory effect of ATP was attenuated by the 5′-nucleotidase (5′-NT) inhibitor AOPCP (0.1 mM) and by the adenosine deaminase (ADA 5 U/mL; Figure 3A), demonstrating that it is dependent on its conversion into adenosine. Interestingly, the inhibitory effect of ATP was reverted into a facilitatory effect in the presence of the NTPDases inhibitor ARL 67156 (0.1 mM; Figure 3A), suggesting that its inhibitory effect may also be dependent on its metabolism into ADP, and that when it is prevented, ATP increases NMDA-evoked [^3^H]-NA release, possibly by activation of P2X receptors.

The characterization of the P2 receptors involved in the inhibitory effect of ATP and ADP indicated that the P2Y_1_ receptors are the main receptors involved, since the inhibitory effect of ATP was partially antagonized by the selective P2Y_1_ antagonist MRS 2500 (1 µM; Figure 3A). Despite the affinity of ADP for the P2Y_1,12,13_ receptor subtypes, its inhibitory effect was not changed by the selective antagonists of the P2Y_12_ and P2Y_13_ subtypes AR-C66096 (10 µM) and MRS 2211 (10 µM), respectively (Figure 3B), being abolished by the P2Y_1_ antagonist MRS 2500 (1 µM; Figure 3B). This group of results suggests that the P2Y_1_ receptor mediates the inhibitory effect of ADP on NMDA-evoked noradrenaline release and contributes to the inhibitory effect of ATP.

### 3.4. Characterization of P2 Receptors Activated by Uracil Nucleotides

UTP and UDP both inhibited NMDA-evoked [^3^H]-NA release (Figure 4), an effect that was abolished by the selective P2Y_6_ receptor antagonist MRS 2578 (1 µM). Furthermore, it was shown that UTP needs to be converted into UDP in order to exert its inhibitory effect, since in the presence of the NTPDases inhibitor ARL 67156 (0.1 mM), UTP effect was abolished (Figure 4). Therefore, inhibition of NMDA-evoked [^3^H]-NA release by uracil nucleotides seems to be mediated by P2Y_6_ receptors.

### 3.5. Effect of Adenosine Receptors on NMDA-Evoked [^3^H]-NA Release

The inhibitory effect of ATP was partially mediated by A_1_ and A_2A_ receptors since it was antagonized by the selective A_1_ antagonist DPCPX and by the selective A_2A_ antagonist SCH 58261 (Figure 3A). To confirm the involvement of A_1_ and A_2A_ receptors on the inhibition of NMDA-evoked [^3^H]-NA release, the effects of the selective A_1_ receptor agonist CPA and the selective A_2A_ receptor agonist CGS 21680 were also investigated. As expected, CPA inhibited the NMDA-evoked [^3^H]-NA release (Figure 5), an effect prevented by the selective A_1_ receptor antagonist DPCPX, but not by the selective A_2A_ receptor antagonist SCH 58261. Interestingly, CGS 21680 also inhibited the NMDA-evoked [^3^H]-NA release, an unexpected effect confirmed by the use of selective antagonists. It was prevented by SCH 58261 but not by DPCPX (Figure 5).

The inhibitory effect mediated by A_2A_ receptors observed is unexpected because these receptors are usually coupled to Gs proteins that activate adenylyl cyclase and protein kinase A (Gs-AC-PKA pathway), leading to an increase in neurotransmitters release [31]. However, the inhibitory effect of the selective A_2A_ receptor agonist CGS 21680, does not involve the Gs-AC-PKA pathway since it was not changed by H-89, a PKA inhibitor (Figure 6).

Studies performed in striatal neurons revealed that adenosine A_2A_ receptors may inhibit the NMDA receptors by activation of a signaling pathway involving calmodulin kinase II (CAMKII) [32]. These results lead us to further explore the involvement of this pathway in the A_2A_ receptor mediated inhibition of NMDA-evoked [^3^H]-NA release in rat brain cortex. The inhibitory effect of CGS 21680 was abolished by the phospholipase C (PLC) inhibitor U-73122, but not changed by its inactive analogue U-73343 (Figure 6).

Additionally, the inhibitory effect of CGS 21680 was also abolished by the protein kinase C (PKC) inhibitor Ro 32-0432 and by the CAMKII inhibitor KN-93 (Figure 6). Taken together, these results suggest that, in rat brain cortex, A_2A_ receptors exert their inhibitory effect on NMDA-evoked [^3^H]-NA release via PLC activation and with the involvement of the enzymes PKC and CAMKII.

## 4. Discussion

The present study demonstrates the existence of a purinergic modulation of noradrenaline release induced by activation of glutamate receptors in the rat brain cortex. The glutamate receptors involved are NMDARs. This conclusion is supported on the inhibition of NMDA-evoked release of noradrenaline caused by the selective NMDA receptor antagonists MK-801 [33] and D-AP5 [34] but not by the antagonist of AMPA and of kainate receptors DNQX [35]. It is further supported by the blockade of NMDA-evoked noradrenaline release upon superfusion with medium containing Mg^2+^, which causes a voltage-dependent block of these receptors [36].

In the present study, the effects of drugs were evaluated by the ratio of [^3^H]-NA release evoked by two periods of stimulation with 30 μM NMDA. The peak of [^3^H]-NA release was very similar in both stimulation periods, indicating that no significant desensitization of NMDARs occurred. Prevention of NMDARs desensitization can be explained by the lower concentration of NMDA used in our experiments than in previous studies [14,15,16]. In addition, the longer interval used between the two periods of stimulation (45 min) might have contributed to the recovery of some NMDARs that could have desensitized.

The mechanism involved in the NMDA-evoked noradrenaline release is accepted to be an exocytotic release dependent on an action potential. The activation and opening of NMDA-gated ion channels, causes an influx of cations and a local membrane depolarization that activates the VSNaC, which spreads the depolarization and activates voltage-dependent calcium channels coupled to the exocytotic release of the neurotransmitter.

In rat brain cortex, several purinergic receptors are involved in the modulation of electrically-evoked noradrenaline release [19]. In general, electrically-evoked neurotransmitter release is a model to study the function of wider neural circuits but is not suitable to investigate the effects on a particular neurotransmitter locally released, such as glutamate. Therefore, it is not known if the glutamate-evoked noradrenaline release is also under modulation by a well-known neuromodulator system such as the purinergic system.

In the present study, we demonstrate that purinergic receptors also modulate noradrenaline release evoked by activation of NMDA receptors. All adenine nucleotides tested (ATP and ADP) inhibited NMDA-evoked noradrenaline release. ATP and ADP are metabolically unstable, being sequentially metabolized by NTPDases and 5′-nucleotidase [37,38] into adenosine, which activates the main inhibitory receptors in the brain, the A_1_ receptors [39]. Under the present experimental conditions, the metabolically generated adenosine seems to contribute to the inhibition of NMDA-evoked noradrenaline release caused by adenine nucleotides, since the inhibitory effect caused by ATP was attenuated in the presence of the 5′-nucleotidase inhibitor AOPCP, which prevents adenosine formation from other nucleotides and in the presence of adenosine deaminase, which metabolizes adenosine into its inactive metabolite. Interestingly, when ATP degradation was prevented upstream to 5′-nucleotidase, by inhibiting NTPDases (therefore preventing ADP formation), a facilitatory effect of ATP emerged indicating that, like in other brain regions [40,41], ATP may also activate P2 receptors (likely of the P2X type), that mediate a facilitation of noradrenaline release. Therefore, the effect of ATP on noradrenaline release evoked by activation of NMDARs is balanced between a facilitation of release, in case enough ATP is preserved from degradation, and an inhibition of noradrenaline release caused by the ATP metabolization products (ADP and adenosine). ADP is the natural agonist of the P2Y_1,12,13_ receptor subtypes, but only the P2Y_1_ receptor seems to contribute to its inhibitory effect on NMDA-evoked noradrenaline release. This conclusion is supported by the observation that the inhibitory effect of ADP was attenuated by MRS 2500, a selective antagonist of P2Y_1_ receptors [42] but not by the selective P2Y_12_ receptor antagonist AR-C 66096 [43] or the selective P2Y_13_ receptor antagonist MRS 2211 [44]. P2Y_1_ receptors have also been shown to inhibit noradrenaline release in other brain regions such as hippocampus [45] and spinal cord [46]. Additionally, they can also modulate NMDARs function in layer V pyramidal neurons of the rat prefrontal and parietal cortex [47].

The uracil nucleotides UTP and UDP present a different selectivity for P2Y receptors: UTP activates mainly P2Y_2,4_ subtypes, whereas UDP is selective for P2Y_6_ receptors [30]. The present results are compatible with the involvement of P2Y_6_ receptors in the uracil nucleotide-induced inhibition of NMDA-evoked noradrenaline release because the UDP effect was abolished by MRS 2578, a selective antagonist of P2Y_6_ receptors [48], and by preventing the conversion of UTP into UDP. The NTPDases inhibitor ARL 67156 prevented the inhibitory effect of UTP. To our knowledge, neither UTP nor UDP are co released with noradrenaline from the noradrenergic neurons but the P2Y_6_ receptors are highly expressed in the rat cerebral cortex [49], mostly in astrocytes and microglia [50,51]. Therefore, UDP modulation of noradrenaline release may have physiological or pathophysiological relevance [52]. Hypoxic/ischaemic conditions may trigger the release of UTP and its metabolic products which may participate in the inflammatory response observed under pathological conditions or in neuroprotection mechanisms induced by NMDRs-mediated excitotoxicity [53]. Although the most likely mechanism to explain the UDP effect is a presynaptic inhibition of noradrenaline release, other mechanisms cannot be excluded since this nucleotide favors noradrenaline uptake by glial cells [19].

The attenuation of the ATP effect by blockade of 5′-nucleotidase points to a contribution of adenosine receptors in the ATP inhibition of NMDA-evoked noradrenaline release. The adenosine receptor most frequently involved in the inhibition of neurotransmitter release is the A_1_ subtype [31]. The present results clearly point to the involvement of A_1_ receptors in the inhibition of NMDA-evoked release of noradrenaline. The effect of ATP was attenuated by DPCPX, a selective antagonist of A_1_ receptors [54]. The selective agonist of A_1_ receptors CPA [55] inhibited NMDA-evoked noradrenaline release as well, an effect also abolished by DPCPX. Additionally, it was found that SCH 5826, a selective antagonist of A_2A_ receptors [56] attenuated the inhibitory effect of ATP. CGS 21680, a selective A_2A_ receptor agonist [57], also decreased NMDA-evoked noradrenaline release, an effect then abolished by SCH 58261. This is an unexpected result because usually the adenosine A_2A_ receptors mediate a facilitation of electrically-evoked neurotransmitter release through activation of the Gs-AC-PKA pathway [58]. Furthermore, the inhibitory effect mediated by A_2A_ receptors on NMDA-evoked noradrenaline release contrasts with our previous results on electrically stimulated cortical brain slices [19], where no effect mediated by these receptors was observed. This discrepancy may be due to a distinct mechanism of the A_2A_ receptor to modulate NMDA-evoked noradrenaline release. Studies carried out in the striatum revealed that A_2A_ receptors may inhibit the conductance of NMDA receptor channels [32,59]. The intracellular signaling of A_2A_ receptor-mediated inhibition of neurotransmitter release involves PLC activation or the activation of PKC and CAMKII, which kinase activity cause inhibition of NMDARs [32,60,61]. The present study indicates that a similar mechanism is involved in the rat brain cortex because the effect of the selective A_2A_ receptor agonist CGS21680 was abolished by inhibition of PLC, PKC and CAMKII.

The results obtained in this study reveal a complex scenario of a purinergic inhibition of NMDA-evoked noradrenaline release in rat brain cortex with a convergence of different pathways (Figure 7). These pathways can be activated by the ATP metabolites (ADP and adenosine), after co-release of ATP with noradrenaline, or by UTP, released mainly by glial cells under partially physiological but mostly pathophysiological conditions.

The cortical organization of noradrenergic and glutamatergic neurons favors this type of interaction. It has been shown that glutamate can diffuse out of the glutamatergic synapse, influencing the activity of neurons in the neighborhood that are not tightly involved in the synapse [62,63]. The noradrenergic synapses are diffuse, whose varicosities and segments between varicosities express receptors (such as extrasynaptic NMDA receptors) that may be easily activated by neurotransmitters such as glutamate and neuromodulators released by any cell type present in the microenvironment [13,62,64].

The purinergic receptors identified as being involved in the inhibition of NMDA-evoked noradrenaline release may represent a significant mechanism modulating NMDA-induced neuroplasticity and providing neuroprotection to noradrenergic circuitry in the cortex.

## 5. Conclusions

The results obtained in this study reveal a complex scenario of a purinergic inhibition of NMDA-evoked noradrenaline release in rat brain cortex with a convergence of different pathways (Figure 7). Purinergic modulation of NMDA-evoked noradrenaline release in rat brain cortex might protect against dysregulated NMDAR signaling observed in several pathologies such as epilepsy, neuropsychiatric disorders, and cognitive dysfunction [27,65,66,67,68]. Future studies are needed to explore whether these purinergic mechanisms are preserved in these pathologies.

Since, in the cortex, noradrenaline has a permissive effect on glutamate-evoked long-term potentiation or long-term depression [8,9], the present results support an involvement of purinergic mechanisms on the modulation of the noradrenaline component of the glutamate mediated neuroplasticity and reveal new targets to pharmacologically modulate NMDA-induced neuroplasticity.

## Figures and Tables

**Figure 1 cells-12-01690-f001:**
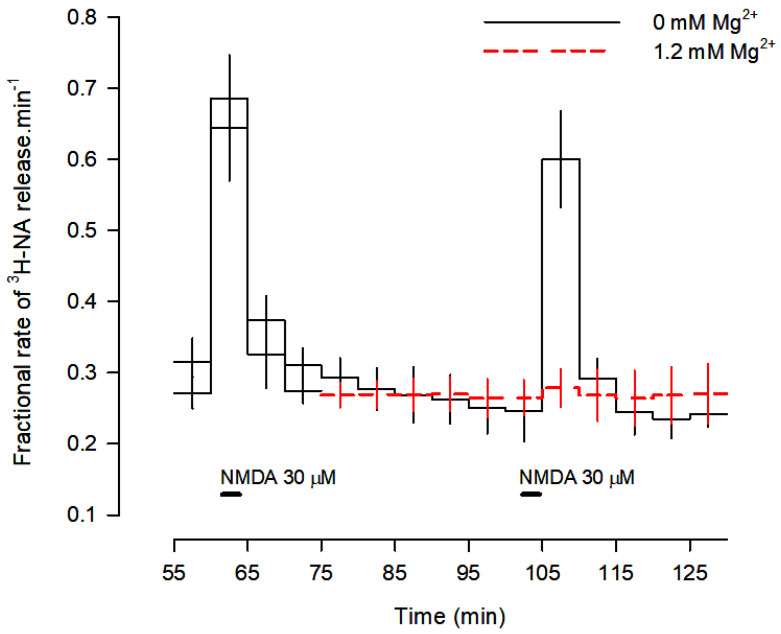
Profile of [^3^H]-NA release evoked by activation of NMDA receptors. Cortical brain slices were incubated with 0.1 μM [^3^H]-NA and superfused at 0.6 mL/min with buffer with and without Mg^2+^ (red dashes). Two periods of stimulation with 30 μM NMDA were applied for 2 min each: S_1_ at 60 min and S_2_ at 105 min of superfusion. Mg^2+^ was added to the superfusion medium 30 min before S_2_ and kept until the end of the experiment (red dashes). Values are means ± SD from five experiments.

**Figure 2 cells-12-01690-f002:**
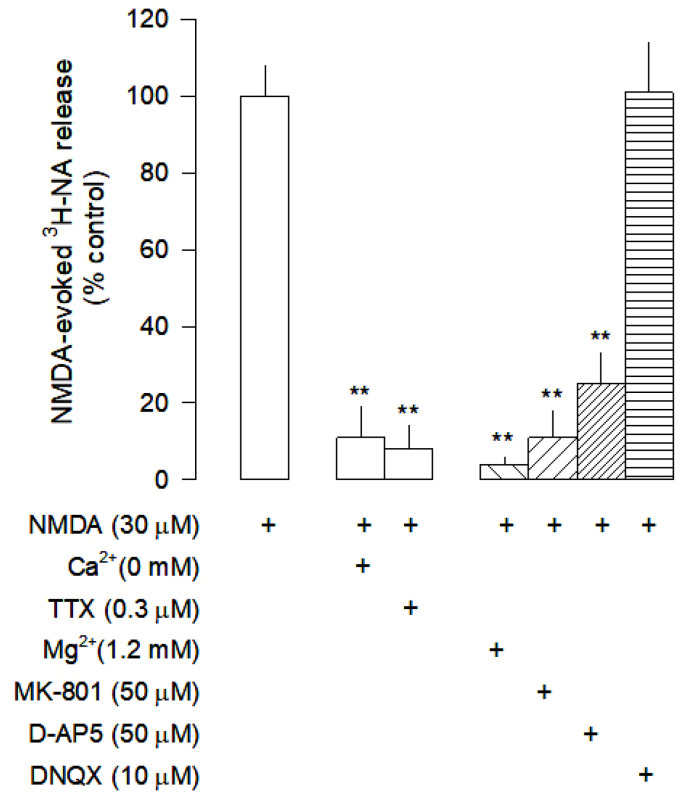
Characterization of NMDA-evoked [^3^H]-NA release from rat cortical slices. The slices were stimulated for two periods (S_1_-S_2_), 45 min apart, with 30 µM NMDA for 2 min. Mg^2+^ was added to the superfusion medium 30 min before S_2_ and kept until the end of the experiment; Ca^2+^ was omitted from the superfusion medium 30 min before S_2_ until the end of the experiment. The inhibitor of VSNaC tetrodotoxin (TTX), the selective antagonists of NMDA receptors MK-801 and D-AP5, and the antagonist of AMPA and kainate receptors DNQX were added to the superfusion medium 20 min before S_2_ and kept until the end of the experiment. The effect of drugs is expressed as % change or % inhibition from NMDA-evoked [^3^H]-NA release; (+) signs placed bellow each column indicates the combination of drugs used under the respective experimental conditions. Values are mean ± SD from five experiments. ** *p* < 0.01, significant different from the effect of NMDA alone.

**Figure 3 cells-12-01690-f003:**
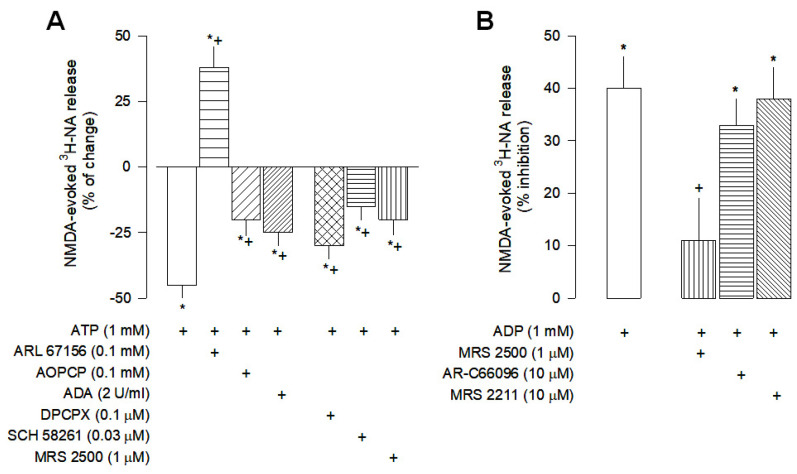
Characterization of ATP (**A**) and ADP (**B**) effects on NMDA-evoked [^3^H]-NA release from rat cortical brain slices. The slices were stimulated for two periods (S_1_-S_2_), 45 min apart, with 30 µM NMDA for 2 min. ATP or ADP were added to the superfusion medium 10 min before S_2_ and kept until the end of the stimulation period. The inhibitor of NTPDases ARL 61756, the 5′-nucleotidase inhibitor AOPCP, the adenosine deaminase (ADA), the selective antagonist of A_2A_ receptors SCH 58261 and the selective antagonists of P2Y_1_, P2Y_12_ and P2Y_13_ receptors, MRS 2500, AR-C66096 and MRS 2211, respectively were added to the superfusion medium 20 min before S_2_ and kept until the end of the experiment. The selective antagonist of A_1_ receptors DPCPX was present from the beginning of superfusion until the end; (+) signs under each column indicates the combination of drugs used in the respective experimental conditions. The effect of drugs is expressed as % change or % inhibition from NMDA-evoked [^3^H]-NA release. Values are mean ± SD from four-six experiments. * *p* < 0.05, significant different from respective control (solvent); ^+^ *p* < 0.05, significant different from the effect of ATP or ADP alone.

**Figure 4 cells-12-01690-f004:**
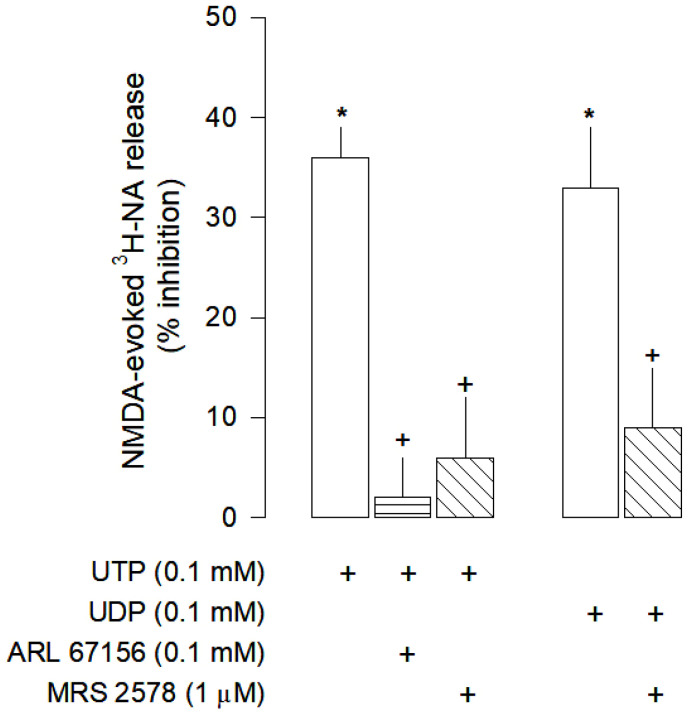
Characterization of UTP and UDP effects on NMDA-evoked [^3^H]-NA release from rat cortical brain slices. The slices were stimulated for two periods (S_1_-S_2_), 45 min apart, with 30 µM NMDA for 2 min. UTP or UDP were added to the superfusion medium 10 min before S_2_ and kept until the end of the stimulation period. The NTPDases inhibitor ARL 61756 and the selective antagonist of P2Y_6_ receptors MRS 2578 were added to the superfusion medium 20 min before S_2_ and kept until the end of the experiment; (+) signs under each column indicates the combination of drugs used in the respective experimental conditions. The effect of drugs is expressed as % inhibition of NMDA-evoked [^3^H]-NA release. Values are mean ± SD from four-six experiments. * *p* < 0.05, significant different from respective control (solvent); ^+^ *p* < 0.05, significant different from the effect of UTP or UDP alone.

**Figure 5 cells-12-01690-f005:**
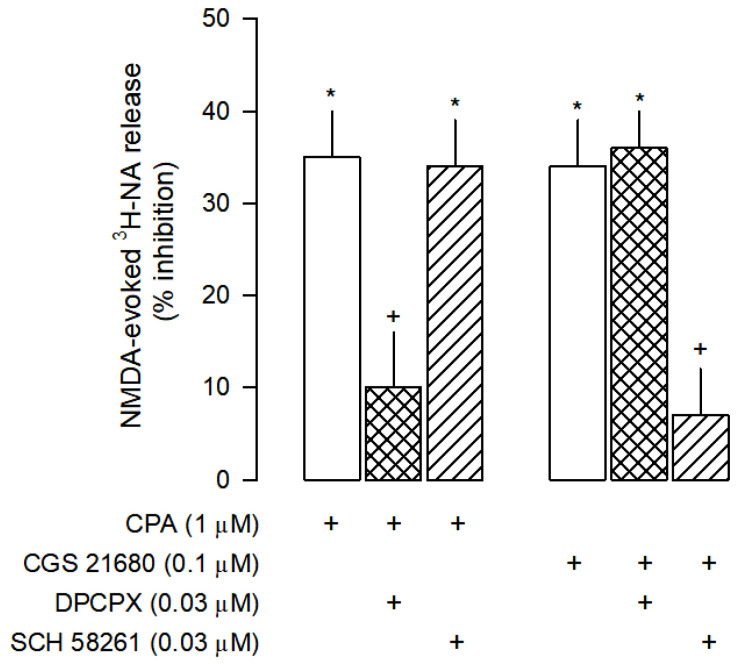
Characterization of A_1_ and A_2A_ receptors effects on NMDA-evoked [3H]-NA release from rat cortical brain slices. The slices were stimulated for two periods (S_1_-S_2_), 45 min apart, with 30 µM NMDA for 2 min. The selective A_1_ agonist CPA and the selective A_2A_ agonist CGS 21680 were added to the superfusion medium 10 min before S_2_ and kept until the end of the stimulation period. The selective antagonists of A_1_ and A_2A_ receptors, DPCPX and SCH 58261, respectively, were added to the superfusion medium 20 min before S_2_ and kept until the end of the experiment; (+) signs under each column indicates the combination of drugs used in the respective experimental conditions. The effect of drugs is expressed as % inhibition of NMDA-evoked [^3^H]-NA release. Values are mean ± SD from six experiments. * *p* < 0.05, significant different from control (solvent); ^+^ *p* < 0.05, significant different from the effect of CPA or CGS 21860 alone.

**Figure 6 cells-12-01690-f006:**
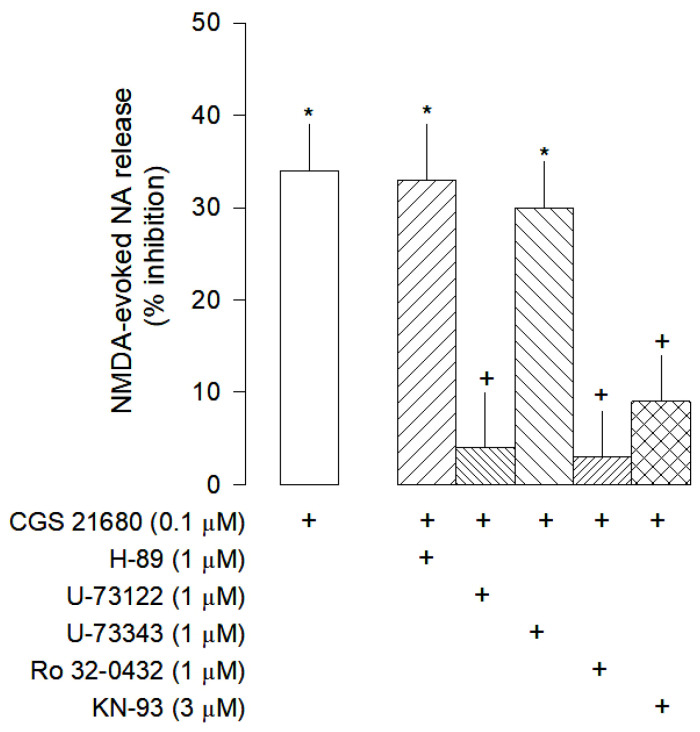
Intracellular signalling pathway of adenosine A_2A_ receptors on the modulation on NMDA-evoked [^3^H]-NA release from rat cortical brain slices. The slices were stimulated for two periods (S_1_-S_2_), 45 min apart, with 30 µM NMDA for 2 min. The selective A_2A_ agonist CGS 21680 were added to the superfusion medium 10 min before S_2_ and kept until the end of the stimulation period. The inhibitors of PKA H-89, of the PLC U-73122 and its inactive analogue U-73343, the inhibitor of PKC Ro 32-0432 and the CAMKII KN-93 were added to the superfusion medium 20 min before S_2_ and kept until the end of the experiment (+) signs under each column indicates the combination of drugs used in the respective experimental conditions. The effect of drugs is expressed as % inhibition of NMDA-evoked [^3^H]-NA. Values are mean ± SD from six experiments. * *p* < 0.05, significant different from control (solvent); ^+^ *p* < 0.05, significant different from the effect of CG 21680 alone.

**Figure 7 cells-12-01690-f007:**
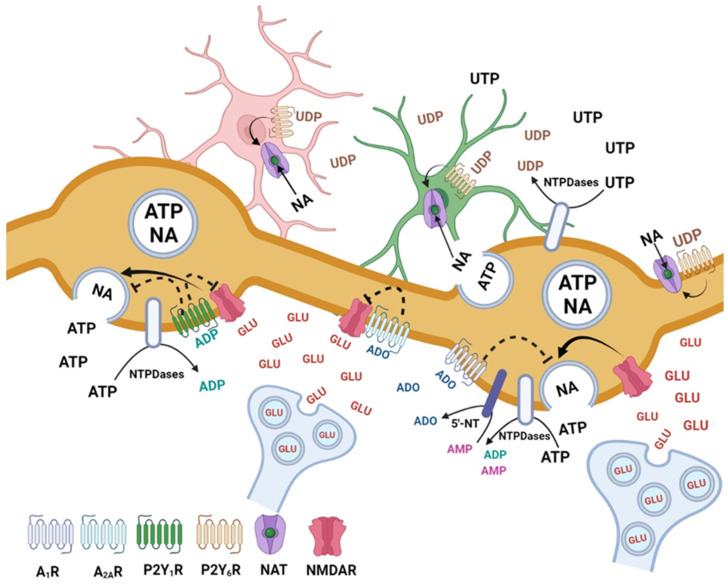
Hypothetical mechanisms involved in the inhibition of adenine and uracil nucleotides of noradrenaline (NA) release evoked by activation of NMDARs, in the rat brain cortex. The NMDARs activated by the spillover of glutamate from glutamatergic synapses stimulate NA release from noradrenergic neurons ATP is co-released with NA from noradrenergic neurons and UTP is released from neurons and glial cells under basal or pathological conditions. ATP and UTP are rapidly metabolized by NTPDases and 5′-nucleotidase giving rise to the metabolites ADP, UDP and adenosine, which are the main compounds involved in the inhibition of NMDARs-evoked NA release. Activation of P2Y_1_, P2Y_6_, A_1_ and A_2A_ receptors may inhibit NMDA-evoked NA release by several mechanisms. The P2Y_1_ (activated by ADP) and the A_1_ and A_2A_ adenosine receptors may act on noradrenergic varicosities and cause inhibition of NA exocytosis. The P2Y_1_ and A_2A_ receptors may also contribute to inhibit the function of NMDARs. The P2Y_6_ receptors expressed mainly by glial cells may increase the uptake of NA contributing, indirectly, to the inhibition of NA release. Figure created with BioRender.com (accessed on 5 May 2023).

**Table 1 cells-12-01690-t001:** Effect of several nucleotides on NMDA-evoked [^3^H]-NA release from rat cortical brain slices.

P2 Receptor Agonists	mM	NMDA-Evoked [^3^H]-NA Release (% Control)
Solvent	--	100 ± 6 (8)
ATP	0.1	71 ± 6 (8) *
ATP	1	59 ± 6 (8) *^+^
ADP	0.1	68 ± 7 (8) *
ADP	1	60 ± 6 (6) *
UTP	0.1	61 ± 7 (6) *
UDP	0.1	67 ± 7 (8) *

The slices were stimulated for two periods (S_1_-S_2_), 45 min apart, with 30 µM NMDA for 2 min. Nucleotides were added to the superfusion medium 10 min before S_2_ until the end of the stimulation period. Results are expressed as % NMDA-evoked [^3^H]-NA release in the absence of drugs (solvent). Values are mean ± SD from (*n*) experiments. * *p* < 0.05, significant different from respective control (solvent). ^+^
*p* < 0.05, significant different from de effect of ATP 0.1 mM.

## Data Availability

The data presented in this study are available on request from the corresponding author.

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
