# Peer review of "Involvement of P2Y1, P2Y6, A1 and A2A Receptors in the Purinergic Inhibition of NMDA-Evoked Noradrenaline Release in the Rat Brain Cortex"

_cells, 2023, doi:10.3390/cells12131690_

Round 1

Reviewer 1 Report

The authors provide interesting and new data, that noradrenaline release in the cortex strongly depends on both NMDA-receptor activation and action potential generation. Significant attenuation of this release was also produced by both ionotropic and metabotropic purinergic receptors activation.

The study has a good experimental design and can be accepted for publication after minor corrections:

(1) In Figure 1 it is easy to add a horizontal bar indicating the presence of Mg2+. It will simplify the understanding of the experimental protocol.

(2) At line 143 "..Mg2+ was added and Ca2+ was omitted.." Probably "..Mg2+ was added or Ca2+ was omitted.." was meant.

Author Response

We thank the Reviewer for his/her comments and suggestions. All points were addressed as explained below.

  • The Figure 1 was changed for better understanding of the experimental protocol of the experiments.
  • In the line 143 changes were made according to the correction suggested that in fact translate what was done.

Reviewer 2 Report

Quintas and coworkers have shown that in the rat cortex, NMDA-evoked noradrenaline release is modulated by several purinergic receptors that may represent a relevant mechanism to regulate the permissive effect of noradrenaline on NMDA-induced neuroplasticity and neurotoxicity. The main aim of the study is interesting, the experimental design and the obtained results are accurate, and the authors have discussed the results appropriately in context. However, I have some minor comments in the method section that need to be improved or addressed before publication.

2.2. Preparation of slices from rat brain cortex

The experimental groups and the number of animals in total and in each experimental group are unknown. How many rats were used in total? Did you have different experimental groups? Or You prepared the brain slices from a group of rats and treated them with different drugs or treatments? The authors must clearly explain these issues in the “animals’ section”.

Figure 1: The difference between the two lines indicating the two treatments (different concentrations of Mg2+) is unclear, especially after printing. I suggest using the thicker line or dashed line for the representation of the second condition in Figure 1.  

In all figures, please use SD instead of SEM. The standard error mean is SD divided by the square root of n (number per group). Therefore, SD shows the distribution of data in each group more precisely.

The quality of the English language is good, but it can be improved. 

Author Response

We thank the Reviewer for his/her comments and suggestions. All points were addressed and text altered accordingly, as explained below

2.2. Preparation of slices from rat brain cortex

We used approximately 50 animals in this study. After quick removal and chilling of the brain, other parts of the animals were utilized in different research projects. Each brain yielded six viable cortical slices. Transverse slices were obtained from the occipital-parietal cortex. We discarded the first superficial slice from each brain, and the subsequent three slices from each hemisphere (totaling six slices per animal) of 400 µm were cut. These slices were then incubated for 45 minutes in a warmed (37°C) and gassed (95% O2 and 5% CO2) Krebs solution. Experimental groups were planned and allocated to superfusion chambers in a manner that optimized the use of animals. Each slice was subjected to a single treatment. In some initial experiments, two chambers were used for control treatment (solvent) to ensure the reproducibility of our protocol.

The suggested clarifications about the preparation of brain slices were introduced in the manuscript.

The Figure 1 was changed, according to your suggestion, for better understanding of the experimental protocol of the experiments.

All figures were replotted with SD instead of SEM as suggested.

Reviewer 3 Report

The manuscript by Quintas et al entitled " Involvement of P2Y1, P2Y6, A1 and A2A receptors in the purinergic inhibition of NMDA-evoked noradrenaline release in the rat brain cortex", is very relevant and important to the field of cortical excitotoxicity. Based on the electrophysiological and noradrenaline (NA) release studies, authors find that NMDA-evoked noradrenaline release, in rat cortical slices, is modulated by several purinergic receptors that may represent a relevant mechanism to regulate the permissive effect of noradrenaline on NMDA-induced neuroplasticity and neurotoxicity. Authors supported their conclusion with several important pharmacological interventions during their study. The study is well designed, and the data presented appropriately. I am of the view that, this manuscript has novel, important data and could be published in the journal Cells with following minor clarification/revision. 

1. While authors proposed that this study is relevant in the pathologies related to glutamate induced excitotoxicity, it is not clear if purinergic inhibition of NMDA mediated NA release is not affected in such pathologies. Authors need to either add supporting data for this or discussion based on previous studies.

2. From the discussion it’s not clear if astrocytes or microglial regulation is actively inhibiting NMDA related excitotoxicity. Authors need to clarify.

Overall the quality of English language is good, but only somewher authors used the complicated sentenses.  

Author Response

We thank the Reviewer for his/her comments and suggestions. All points were addressed and text altered accordingly, as explained below.

1. While authors proposed that this study is relevant in the pathologies related to glutamate induced excitotoxicity, it is not clear if purinergic inhibition of NMDA mediated NA release is not affected in such pathologies. Authors need to either add supporting data for this or discussion based on previous studies.

To the best of our knowledge, no studies have been conducted on the purinergic modulation of noradrenaline release evoked by NMDA or other stimuli in disease models. The reviewer is entirely correct; we cannot guarantee that the purinergic mechanisms referenced remain operational in the pathologies mentioned. Therefore, we have revised the text to account for this uncertainty.

2. From the discussion it’s not clear if astrocytes or microglial regulation is actively inhibiting NMDA related excitotoxicity. Authors need to clarify.

Although it is known that UTP released from astrocytes seem to play a global protective activity (see ref 53) we agree with the Reviewer that extrapolation should not be performed to avoid the interpretation that a protective effect of purinergic modulation against NMDA-related excitotoxicity is established. Consequently, we modified the text.